# Identification and Characterization of Antioxidant Enzyme Genes in Parasitoid *Aphelinus asychis* (Hymenoptera: Aphelinidae) and Expression Profiling Analysis under Temperature Stress

**DOI:** 10.3390/insects13050447

**Published:** 2022-05-09

**Authors:** Xiang Liu, Zhi-Xiao Fu, Zhi-Wei Kang, Hao Li, Tong-Xian Liu, Dun Wang

**Affiliations:** 1State Key Laboratory of Crop Stress Biology for Arid Areas, College of Plant Protection, Northwest A&F University, Xianyang 712100, China; liuxiang_edu@163.com; 2Laboratory of Insect Ecology and Molecular Biology, College of Plant Health and Medicine, Qingdao Agriculture University, Qingdao 266109, China; f17853470760@163.com; 3School of Life Science, Institutes of Life Science and Green Development, Hebei University, Baoding 071002, China; zwkang2010@126.com; 4Chengyang District Education and Sports Bureau, Qingdao 266112, China; lamaaralee@163.com

**Keywords:** *Aphelinus asychis*, antioxidant enzymes, thermal stress, gene expression pattern

## Abstract

**Simple Summary:**

High temperature affects the control efficiency of *Aphelinus asychis*, an important parasitic natural enemy of aphids. Antioxidative enzymes can protect organisms against oxidative damage by eliminating excess reactive oxygen species (ROS). This study identified 14 genes belonging to four classes of antioxidant enzymes, including superoxide dismutase (SOD), catalase (CAT), peroxidases (POD), and glutathione-S-transferase (GST). The results showed that the expression levels and enzymatic activities of most antioxidant enzyme genes are significantly induced by high temperature, which indicates that antioxidant enzymes increase the resistance of *A. asychis* to lethal high temperature. Taken together, this study enriches the understanding of the molecular mechanisms of resistance of *A. asychis* to environmental high temperatures.

**Abstract:**

It is well known that high temperature, a typically negative environmental factor, reduces the parasitism of a parasitoid. Generally, high temperature causes the rapid overproduction of reactive oxygen species (ROS) in organisms, and antioxidative enzymes participate in the process of resisting environmental stress by eliminating excess ROS. In the present study, we identify two superoxide dismutase (SOD), one catalase (CAT), six peroxidases (POD), and five glutathione-S-transferase (GST) genes; and the survival rate and antioxidative enzyme patterns under short-term high temperature exposure of the parasitoid wasp, *A.asychis,* are examined. Survival results of *A.asychis* reveal that females show higher thermal tolerance than males. Under short-term high-temperature exposure, in females, the expression levels of most antioxidant enzyme genes decreased first and then increased to a peak at 41 °C, while only the expression of *AasyGST4* showed a continuous increase. In males, the expression patterns of most antioxidant enzyme genes fluctuated and reached a maximum at 41 °C. Moreover, the expression levels of the majority of antioxidant enzyme genes were higher in females than in males. In addition, at temperatures of and above 35 °C, the activities of these four antioxidant enzymes were induced. The results show that the antioxidant enzymes confer thermo-tolerance to *A. asychis* against lethal thermal stress. Our observations enrich the understanding of the response mechanism to high-temperature assaults of *A. asychis*.

## 1. Introduction

Reactive oxygen species (ROS) are metabolites produced in the process of oxidative metabolism in organisms, which mainly include O_2_^−^, H_2_O_2_, OH^−^, and singlet oxygen [1]. ROS mainly originate from the electron transport chain of cell microsomes and mitochondria and participate in the embryonic development, cell differentiation, proliferation, growth, and development of organisms, and are also necessary to parts of cell signaling [2]. Only a small part of the ROS is eliminated by dietary antioxidants such as ascorbic acid and carotenoids, and most are eliminated by a set of antioxidant enzymes [3]. In order to protect themselves from excessive ROS, insects have evolved a complex network of enzymatic antioxidant systems, including enzymes and non-enzymatic components [4]. The antioxidant enzyme system is an important physiological system for resisting oxidative stress in insects, including superoxide dismutase (SOD), catalase (CAT), peroxidase (POD), and glutathione-S-transferase (GST) [5]. SOD catalyzes the disproportionation reaction of superoxide anions to generate hydrogen peroxide and molecular oxygen, while CAT and POD work together to decompose hydrogen peroxide into water and oxygen [6,7,8]. GST can eliminate the toxic products of lipid peroxidation in cells [9,10].

Insects are ectotherm animals, and their physiological functions, behaviors, and population fitness are affected by temperature, especially abnormally high or low temperatures, which affect the fitness of insect populations, and thereby further impact population dynamics [11,12]. Under normal conditions, the production and elimination of ROS are in a dynamic balance. However, high-temperature stress induces a large accumulation of cellular ROS, leading to lipid peroxidation; damage to cell membrane fluidity; cell apoptosis; and DNA damage in the form of mutations, base deletions, degradation, and single-strand breaks [13].

*Aphelinus asychis* Walker (Hymenoptera: Aphelinidae) is a widely distributed endoparasitic wasp that parasitizes more than 60 species of aphids and has been widely used in the field of biological control for a variety of aphids [14,15]. Generally, the temperature shows irregular changes in day and night cycles, and exceeds 40 °C in certain periods in the greenhouse and field (Climate Databases, Chinese Academy of Forestry). Short-term high temperature affects the physiological state of natural enemies [16,17], so it has become a key factor in restricting the use of parasitic wasps. Since high temperature significantly reduces the survival and development time of *A. asychis*, and affects the control efficiency of progeny [18,19], it is necessary to explore the molecular mechanism of *A. asychis* responding to high temperature stress. Although the functional expression patterns of antioxidant enzymes in response to different types of external environmental stresses are determined in many insects [5,20,21,22], there is a lack of information about antioxidant enzymes in parasitoid wasps, an important biological control of natural enemies.

In the present work, based on transcriptome data, we assess four antioxidant enzyme family genes, including two SOD, one CAT, six POD, and five GST genes, and evaluate their expression patterns and enzyme activity changes under different temperature treatments. In addition, the genetic structure and evolutionary characteristics of these identified antioxidant enzyme genes are also analyzed. Our results not only reveal the potential role of antioxidant enzyme genes in response to short-term high-temperature stress, but also provide valuable information to both the rearing and releasing of *A. asychis*.

## 2. Materials and Methods

### 2.1. Insect Species

The *A**. asychis* colony was originally collected from *Myzus persicae* (Sulzer) in Chinses cabbage, *Brassica pekinensis*, at the Experimental Farm of Northwest A&F University (34°17′37.01″ N, 108°01′03.34″ E). The laboratory population was reared in mesh-covered cages (40 × 40 × 40 cm). Six pots of fresh winter wheat seedlings (*Triticum aestivum* L. Var. “Xinong 979”) with *Sitobion avenae* (Fabricius) were placed in each cage. The environmental parameters of the air-conditioned insectary were set at 25 ± 1 °C, a photoperiod of 16L: 8D, and relative humidity of 60 ± 5%.

### 2.2. Temperature Treatments

To characterize the expression level and enzyme activity of antioxidant enzymes in *A.*
*asychis* under short-term high-temperature stress, 100 newly emerged adults (1-day-old) were put into a Petri dish (12.0 cm in diameter and 2.0 cm in height) as a group for one hour of heat shock. Based on the temperature detected in the greenhouse, five temperature treatments were set (25 °C—control, 32.5, 35, 37.5, 40, and 41 °C). The survival rates were counted after all temperature-treated groups were resuscitated at 25 °C for one hour and the surviving intact adults were flash-frozen with liquid nitrogen immediately and stored in the −80°C refrigerator until used for RNA extraction. Each treatment had three biological replicates.

### 2.3. Identification of the Antioxidant Enzyme Genes in A. asychis

Based on the functional annotation of *A. asychis* transcriptome data, we identified candidate antioxidant enzyme genes. The amino acid sequences of all the candidate genes obtained were imported into the National Center for Biotechnology Information (NCBI) Conserved Domain Database (http://www.ncbi.nlm.nih.gov/cdd, accessed on 5 April 2022) [23] to confirm whether there was a conserved domain of the family protein. The isoelectric point and molecular weight were calculated by inputting the amino acid sequence of the antioxidant enzyme into pI/Mw from ExPASy (http://web.expasy.org/cgibin/compute_pi/pi_tool, accessed on 5 April 2022) [24]. The subcellular localization was predicted by inputting the amino acid sequence of the antioxidant enzyme into CELLO v2.5 (http://cello.life.nctu.edu.tw/, accessed on 5 April 2022) [25].

### 2.4. Phylogenetic and Structural Feature Analysis of Antioxidant Enzymes Genes

The amino acid sequences of all identified antioxidant enzymes and other amino acid sequences used to construct the phylogenetic tree were aligned by MAFFT multiple alignment from GenomeNet (https://www.genome.jp/tools-bin/mafft, accessed on 5 April 2022) [26] with the default option. Subsequently, the phylogenetic trees were constructed using a neighbor-joining (NJ) method in the software MEGA software (version 5.2) [27]. The method parameters were set as follows: Poisson correction model, pairwise deletion. Bootstrap analysis with 1000 replicates was used to evaluate the reliability of the internal branches. The trees were further optimized using iTOL tool (https://itol.embl.de/, accessed on 5 April 2022). In addition, the secondary structures (α-helix and β-sheet) of antioxidant enzyme proteins were predicted by inputting the amino acid sequence of the antioxidant enzyme into ESPript 3.0 (https://espript.ibcp.fr/ESPript/cgi-bin/ESPript.cgi, accessed on 5 April 2022) [28].

### 2.5. RNA Extraction and Expression Profiles of Antioxidant Enzyme Genes

The samples stored at −80 °C were ground in TissueLyser after adding TRIzol reagent (Takara Bio, Tokyo, Japan) to extract RNA. The total RNA extracted was dissolved in an appropriate amount of RNase-free water, and the concentration and quality were detected by a NanoDrop ND-2000 spectrophotometer. Then, a total of 1000 ng of RNA per system was reverse transcribed into cDNA using a PrimeScript™ RT reagent Kit with gDNA Eraser (perfect real time) (Takara, Dalian, China). The relative expression of related genes was determined on the iQ™ 5 Multicolor Real-Time PCR Detection System (Bio-Rad, Hercules, CA, USA), using the gene-specific primers designed by the NCBI primer design online tool (https://www.ncbi.nlm.nih.gov/tools/primer-blast/index.cgi?LINK_LOC=BlastHome, accessed on 5 April 2022), as shown in Appendix A. The RT-qPCR reaction system consists of: 10 μL of TB Green® Premix Ex TaqTM II (Tli RNaseH Plus) (Takara Bio, Tokyo, Japan), 0.8 μL of forward and reverse primers (10 mM), 0.8 μL of sample cDNA, and 7.6 μL of sterilized ultra-pure grade H_2_O. Cycling conditions were 95 °C for 30 s, 40 cycles of 95 °C for 5 s, and 58 °C for 30 s. In each RT-qPCR experiment, each gene was run three times from three biological replicates as three technical replicates. Relative quantification was performed using the Comparative 2^−^^ΔΔ^^CT^ method. The *18s RNA* gene (forward primer 5′-3′: GGGAATCGTATCCGTGGACC; reverse primer 5′-3′: GCTCGTGGGTCGATGAAGAA) was used as the reference gene to normalize the transcription data. In addition, reference genes were evaluated in each RT-qPCR.

### 2.6. Enzyme Activity Assay

Commercially available assay kits (Nanjing Jiancheng Bioengineering Institute, Jiangsu, China) were used to detect changes in the activity of antioxidant enzymes (SOD, CAT, POD, and GST) under different temperature treatments. Sample preparation and the method of use were as previously described [29].

### 2.7. Statistical Analyses

The experimental data were expressed as mean ± standard error, and the statistical analysis was performed using IBM SPSS 22.0 software (SPSS Inc., Chicago, IL, USA). For the analysis of statistical differences between two and multiple groups, the Student’s *t*-test and one-way analysis of variance (ANOVA) were used, respectively. The Fisher’s protected least significant difference (LSD) test was used to adjust the separation of means. The significance levels of these tests were set at *p* < 0.05.

## 3. Results

### 3.1. Effect of Short-Term High Temperate on the Survival Rate of A. asychis

The effects of short-term heat shock on the survival rate of *A. asychis* adults are shown in Figure 1. Temperature treatment below 35 °C did not significantly reduce the survival rate of *A. asychis* (Figure 1). However, when the temperature increased from 35 °C to 41 °C, the survival rate of female adults and male adults decreased significantly, from 92.2% to 17.3% and 89.7% to 4.6%, respectively. At 37.5, 40, and 41 °C, the survival rates of female adults were significantly higher than those of male adults. In general, short-term high temperature stress reduced the survival rate of *A. asychis*, and compared to male adults, female adults had greater resistance to high-temperature stress.

### 3.2. The Identification and Phylogenetic Relationship Analysis of Antioxidant Enzyme Genes in A. asychis

Based on transcriptome data, 14 antioxidant enzyme genes were identified (Table 1), and the predicted proteins of the identified antioxidant enzyme genes were preliminarily classified according to the NCBI Conserved Domain Database (NCBI CDD), including two SOD, one CAT, six POD, and five GST genes.

The antioxidant family genes detected in this study were compared with those in the NCBI database. All had >50% identity with all antioxidant family genes and the POD family was the largest enzyme family detected in *A. asychis* (Table 1). The nucleic acid length of the antioxidant enzymes of *A. asychis* ranged from 507 (*AasyPOD1*) to 2325 (*AasyPOD6*). The majority of the antioxidant enzymes were located in the cytoplasm whereas only *AasyCAT1*, *Aasy**SOD1, AasyPOD3**, Aasy**POD6*, and *AasyGST4* are distributed in organelles such as mitochondria, nucleus, and extracellular (Table 2). Detailed information of the identified antioxidant enzyme genes has been uploaded to the NCBI database.

The phylogenetic analyses of antioxidant enzyme genes of *A. asychis* showed a closer evolutionary relationship with the related genes of *Copidosoma floridanum*, *Nasonia vitripennis* and *Ceratosolen solmsi marchali* (Figure 2A). AasyCAT1 was closely related to CAT in *C. floridanum* (Figure 2A). AasySOD1, AasySOD2, AasyGST3, AasyPOD3 and AasyPOD5 were clustered with related genes in *N. vitripennis* (Figure 2B–D). AasyPOD2 and AasyPOD6 exhibited close relationship with POD in *C. solmsi marchali* (Figure 2C). In addition, five GST genes were categorized as three different classes: AasyGST1 belongs to Class III, AasyGST2, AasyGST3, and AasyGST4 belong to Class I class, and AasyGST5 belongs to Class II (Figure 2B).

### 3.3. Differential Expression of Antioxidant Enzyme Genes in A. asychis under High-Temperature Stress

The expression patterns of antioxidant enzyme genes of *A. asychis* under external temperature stress were investigated using RT-qPCR (Figure 3). In females, the expression levels of all antioxidant enzyme genes showed a trend of first decreasing and then increasing, but the expression of *AasyGST4* continued to increase. In males, the expression of most antioxidant enzyme genes reached their peak at 41 °C, except for *AasyPOD**5*, *AasyPOD**6*, and *AasySOD2*, which peaked at 35 °C. In addition, at 40 °C and 41 °C, the expression levels of all antioxidant enzymes in females were significantly higher than those in males.

### 3.4. Antioxidant Enzyme Activities of A. asychis in Response to High-Temperature Stress

Antioxidant enzyme activities (SOD, CAT, POD, and GST) of *A. asychis* under different temperature treatments were determined, and activities of SOD, CAT, POD, and GST continued to increase with the increase in treatment temperature (Figure 4). GST activity in males is irregular, and peaks at 35 °C and 41 °C. The SOD activity in males increased first and then decreased, and the activity peaked at 35 °C. In addition, when the treatment temperature is less than or equal to 37.5 °C, all antioxidant enzyme activities in males are significantly higher than those in females, and when the temperature is higher than 37.5 °C, the opposite is true.

## 4. Discussions

The most direct impact of environmental high-temperature stress on insect populations is the survival rate of the insects. According to the response of survival rates at high temperatures, the temperature range that is not suitable for development is divided into high lethal temperature and sublethal temperature [30,31]. Under temperature stress, the oxidative stress mechanism in insects is activated, and the activity of the antioxidant enzyme system is rapidly enhanced to remove excess ROS, which reduces the damage caused by oxidative stress [32,33,34,35]. Although antioxidant defense systems (SOD, CAT, POD, and GST) are important for insects to resist damage caused by external adverse environmental conditions; so far, little is known about the mechanism by which the antioxidant enzyme genes of *A. asychis*, an important aphid endoparasitoid, respond to external stresses.

In this work, we conducted five different temperature treatments on female and male adults of *A. asychis*. The results show that *A. asychis* maintain a high survival rate under the heat shock treatment at 35 °C. However, the survival of both female and male adults decrease significantly with increasing treatment temperature. Moreover, compared with males, females show better resistance to high-temperature stress. The phenomenon that high-temperature environment reduces the survival of insects has been reported frequently. For example, the survival rate of various states of *Liriomyza huidobrensis* Blanchard under abnormal high temperature is significantly reduced [36], and the survival rate of adults of *Aphis gossypii* Glover under high-temperature stress is also significantly reduced [37]. The phenomenon that female adults have better heat resistance is also found in another natural enemy of aphids, *Aphidius gifuensis* Ashmead [20]. A reasonable explanation is that the expression of heat shock proteins in the ovaries and embryos of females is at a high level [38,39,40]. The females of *A. asychis* are larger than males [41], which may be another reason why females process higher heat resistance than males do.

At the physiological level, the first reaction of insects after environmental temperature stress is oxidative stress, which causes lipid peroxidation and protein oxidation, leading to cell death [42]. Insects have specific physiological characteristics. The oxygen transported by the tracheal system reaches the tissues through diffusion. During the flight of insects, the tissues are in a strong oxidation state. Plants ingested by phytophagous insects are rich in redox compounds, which increase the body’s oxidative stress under normal conditions. In response to this, insects have developed efficient antioxidant systems (antioxidant enzymes and non-enzymatic antioxidants) to reduce oxidative stress [4].

SOD exists in almost all biological cells. It is the most important enzyme in the organism for working against oxidative stress, and plays an indispensable role in the physiological process of removing high concentrations of superoxide free radicals caused by extreme temperatures [43,44]. In this work, the expression patterns of *AasySOD1* and *AasySOD2* in females and males are different. In females, compared with the control, the expression levels of *AasySOD1* and *AasySOD2* were significantly induced by a temperature greater than 40 °C. In males, the expression of *AasySOD1* was kept at a low level, and the expression of *AasySOD2* reached its peak at 35 °C, and then continued to decrease as the treatment temperature increased. Correspondingly, the activity of SOD continued to increase with the increase in treatment temperature. It was also found that high temperature stress induces an increase in SOD activity in *Bactrocera dorsalis* [45]. As the first step in the antioxidant system, SOD first functions to scavenge excess superoxide anion free radicals [46]. For the expression trend of *SOD2* expression in male adults that first increased and then decreased, we conjectured that the high temperature induced SOD activity to increase in the initial stage in order to eliminate free radicals. We speculated that high temperature induces SOD activity to increase in the initial stage in order to eliminate free radicals, but when the ambient temperature exceeds its tolerance threshold, the metabolic function of insects is destroyed, which affects the expression of *SOD* at the mRNA level.

POD and CAT function synergistically to eliminate H_2_O_2_ generated by SOD decomposing superoxide anion free radicals. The main function of POD is to eliminate low concentrations of H_2_O_2_, while CAT mainly removes high concentrations of H_2_O_2_ [7,47]. In the present study, the activity changes of POD and CAT tended to be the same with increasing temperature. The activities of the above two enzymes in females were higher than those in males, which basically corresponded to the expression profiles of the two enzymes. However, the activities of males were significantly higher than those of females at 35 °C, which needed further study. The POD activity in the larvae of *Tetrastichus brontispae* was decreased with the increase in the treatment temperature, and the activity reached the maximum value at 33 °C [48]. In *Chilo suppressalis*, CAT not only responded to high temperatures greater than 30 °C, but also exposure to pesticides, such as avermectin and chlorantraniliprole, significantly increased *CAT* mRNA levels and enzymatic activities [49].

The main role of GST is to reduce the damage of ROS by metabolizing lipid peroxides [50]. At 41 °C, the activity of GST reached a maximum in both females and males. In addition, GST activity initially decreased. At the mRNA level, the above phenomenon appeared in the expression of *AasyGST1*, *AasyGST2*, *AasyGST3*, and *AasyGST5*. The expression level of *AasyGST4* in females increased significantly with treatment temperature, while it was stable in males. Due to the lack of genome, we are not certain about whether *AasyGST4* is a nuclear isomer or not. We hypothesized that in males, *AasyGST4* does not seem to be involved in the response to high temperature stress. Based on the above discussion, we speculate that because the defense order of GST is lower than that of SOD and CAT, *A. asychis* may preferentially synthesize enzymes with a higher defense order through transcriptional and translational regulation, which inhibits GST activity.

To summarize, in this study, it is indisputable that both the expression levels and activities of antioxidant enzyme genes are induced by high temperature, and after a short heat shock, the expression of most antioxidant enzyme genes and the activities of all enzymes were higher in females than in males, which is consistent with the result that female adults have better heat resistance than male adults in this experiment, indicating that antioxidant enzymes play a key role in the process of *A. asychis* resistance to high-temperature damage.

## 5. Conclusions

In conclusion, the activities of the four enzymes were determined after temperature gradient treatment, and 14 antioxidant enzyme genes belonging to the four enzymes were identified and characterized. Our study clarified the response mode of the antioxidant enzyme system in *A. asychis* to high-temperature stress, and the significance of heat tolerance of the different sexes of *A. asychis* was analyzed from the perspective of biological phenotype, which provided a molecular basis for further application of *A. asychis*, an important natural enemy of aphids, in greenhouses.

## Figures and Tables

**Figure 1 insects-13-00447-f001:**
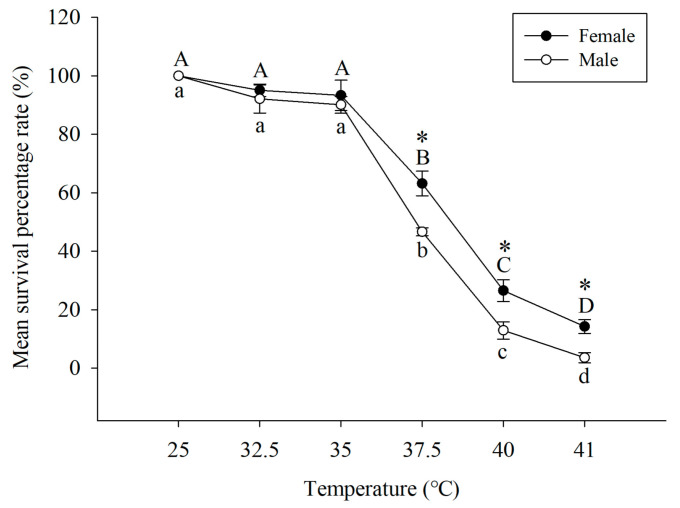
The average survival rate of *Aphelinus asychis*. Uppercase and lowercase letters indicate significant differences between adult females and males treated at different temperatures (*p* < 0.05). Asterisk (*) indicates a significant difference between adult males and females in each temperature treatment (*p* < 0.05).

**Figure 2 insects-13-00447-f002:**
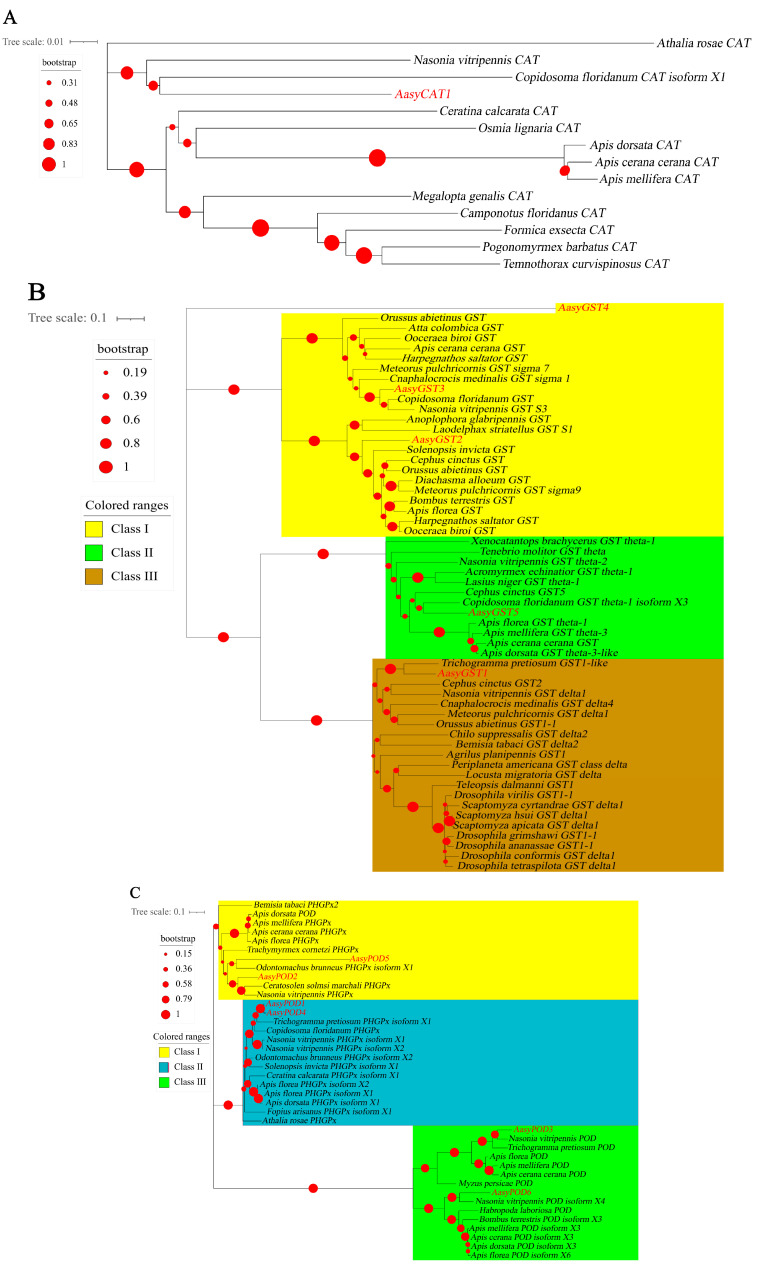
Phylogenetic analysis of antioxidant enzymes in insects: (**A**) catalase; (**B**) glutathione-S-transferases; (**C**) peroxidases; (**D**) superoxide dismutase.

**Figure 3 insects-13-00447-f003:**
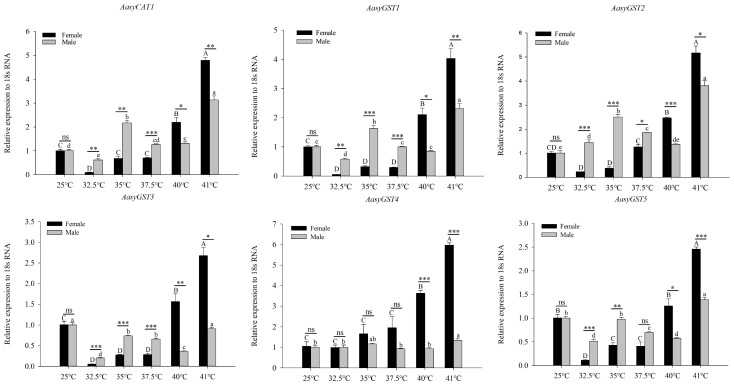
The effect of short-term heat treatment on the relative expression of *Aphelinus asychis* antioxidant enzyme genes. Each value represents the mean (±SE) of three replications. Uppercase and lowercase letters indicate significant differences between adult females and males treated at different temperatures (*p* < 0.05). “*”, “**” and “***” mean significant differences between female and male adults at each temperature treatment, respectively (*p* < 0.05, *p* < 0.01 and *p* < 0.001); “ns” means no significant differences between two columns at each temperature treatment.

**Figure 4 insects-13-00447-f004:**
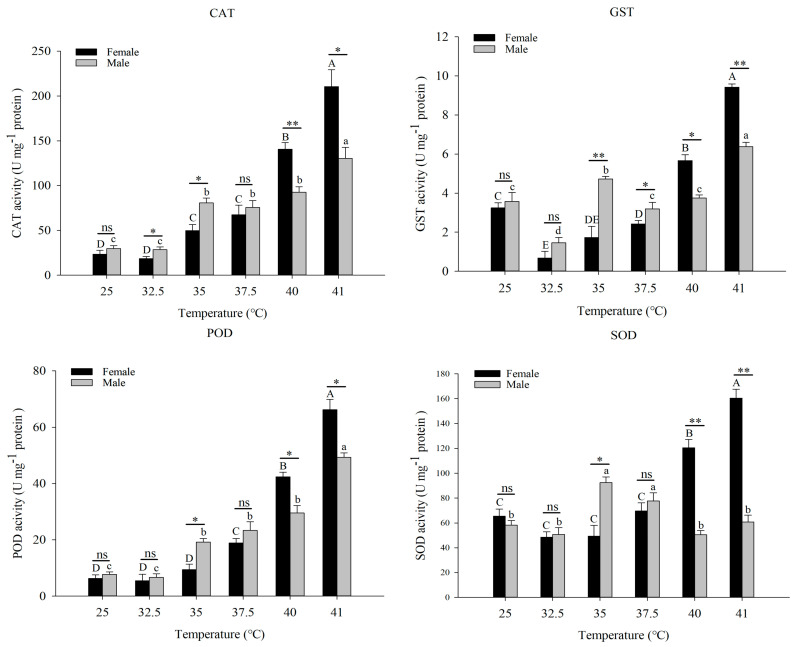
The effect of short-term heat treatment on the antioxidative enzyme activities of *Aphelinus asychis*. Each value represents the mean (±SE) of three replications. Uppercase and lowercase letters indicate significant differences between adult females and males treated at different temperatures (*p* < 0.05). “*”, “**” mean significant differences between female and male adults at each temperature treatment, respectively (*p* < 0.05 and *p* < 0.01); “ns” means no significant differences between two columns at each temperature treatment.

**Table 1 insects-13-00447-t001:** The identified antioxidant enzyme-related genes in *A. asychis*.

	Gene Name	Accession Number	FPKM	Blast P Hit	E-Value	Identity (%)
Catalase	*AasyCAT1*	OK169278	118.72	XP_031785372.1 catalase [*Nasonia vitripennis*]	0.0	84.28%
Superoxide dismutase	*AasySOD1*	OK169290	54.94	KMQ95190.1 superoxide dismutase [*Lasius niger*]	3 × 10^−65^	67.10%
	*AasySOD2*	OK169291	36.33	XP_031787604.1 superoxide dismutase [Cu-Zn] isoform X3 [*Nasonia vitripennis*]	3 × 10^−94^	89.54%
Peroxidase	*AasyPOD1*	OK169284	14.12	XP_008210669.1 probable phospholipid hydroperoxide glutathione peroxidase isoform X1 [*Nasonia vitripennis*]	1 × 10^−109^	90.48%
	*AasyPOD2*	OK169285	299.22	XP_011503666.1 PREDICTED: phospholipid hydroperoxide glutathione peroxidase [*Ceratosolen solmsi marchali*]	3 × 10^−94^	80.36%
	*AasyPOD3*	OK169286	4.57	XP_008203489.1 peroxidase [*Nasonia* *vitripennis*]	0.0	77.03%
	*AasyPOD4*	OK169287	35.98	XP_008210672.1 probable phospholipid hydroperoxide glutathione peroxidase isoform X2 [*Nasonia vitripennis*]	9 × 10^−110^	90.48%
	*AasyPOD5*	OK169288	27.94	XP_001606751.1 phospholipid hydroperoxide glutathione peroxidase-like [*Nasonia* *vitripennis*]	6 × 10^−57^	52.97%
	*AasyPOD6*	OK169289	7.55	XP_008203493.1 peroxidase isoform X4 [*Nasonia vitripennis*]	0.0	70.03%
Glutathione	*AasyGST1*	OK169279	12.59	XP_014225564.1 glutathione-S-transferase 1-like [*Trichogramma pretiosum*]	3 × 10^−121^	78.24%
	*AasyGST2*	OK169280	34.36	XP_031783860.1 glutathione-S-transferase isoform X1 [*Nasonia vitripennis*]	8 × 10^−122^	88.78%
	*AasyGST3*	OK169281	135.33	NP_001165920.1 glutathione-S-transferase S3 [*Nasonia vitripennis*]	5 × 10^−133^	88.61%
	*AasyGST4*	OK169282	19.78	OXU29987.1 hypothetical protein TSAR_001104 [*Trichomalopsis sarcophagae*]	3 × 10^−44^	50.61%
	*AasyGST5*	OK169283	45.22	XP_014215774.1 glutathione-S-transferase theta-1 isoform X3 [*Copidosoma floridanum*]	7 × 10^−124^	73.68%

**Table 2 insects-13-00447-t002:** Information on antioxidant enzyme genes in *A. asychis*.

Family	Gene Name	Coding Sequence	Mw (kDa)	pI	Subcelluar Location	Strand
Catalase	*AasyCAT1*	1818	67.8	9.00	Mitochondrial	plus
Superoxide dismutase	*AasySOD1*	531	18.5	6.29	Extracellular	minus
	*AasySOD2*	675	23.4	5.88	Cytoplasmic	plus
Peroxidase	*AasyPOD1*	507	19.1	6.90	Cytoplasmic	minus
	*AasyPOD2*	579	21.7	6.44	Cytoplasmic	minus
	*AasyPOD3*	2115	79.4	6.13	Nuclear	minus
	*AasyPOD4*	576	21.6	8.61	Cytoplasmic	minus
	*AasyPOD5*	552	20.0	5.37	Cytoplasmic	plus
	*AasyPOD6*	2325	86.3	7.78	Extracellular	minus
Glutathione-S-transferase	*AasyGST1*	654	25.0	6.63	Cytoplasmic	plus
	*AasyGST2*	624	23.8	6.45	Cytoplasmic	plus
	*AasyGST3*	609	23.0	6.00	Cytoplasmic	minus
	*AasyGST4*	1014	38.6	11.74	Nuclear	minus
	*AasyGST5*	684	26.6	7.01	Cytoplasmic	plus

## Data Availability

All the individual data are gathered in the Appendix A.

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
