# Peer review of "Identification and Characterization of Antioxidant Enzyme Genes in Parasitoid Aphelinus asychis (Hymenoptera: Aphelinidae) and Expression Profiling Analysis under Temperature Stress"

_insects, 2022, doi:10.3390/insects13050447_

Round 1

Reviewer 1 Report

The manuscript provides a meaningful contribution to the understanding of climate change at the level of insect parasitism.  The study focuses on identifying and characterizing major antioxidant gene families and antioxidant enzyme activity in the parasitic wasp, Aphelinus asychis, a natural enemy of aphids. increasing temperatures on a parasitic natural enemy of aphids.  The authors generally found increasing levels of antioxidant gene expression and enzyme activity in males and female as temperature increased, however females showed a greater response to heat stress than males. This is a timely paper that will provide baseline data for further studies of metabolic and temperature stress on the parasitic interactions between insects and natural predators.

  • I have to commend the authors on an organized and easily comprehensible manuscript. The introductory material is complete, informative and provides in depth information on available on metabolic stress caused by increased temperatures in insects.. This can be a misinterpreted and confusing topic and the authors explained it very concisely and clearly. The author’s use of the literature is very good.  To my knowledge this work has not been published elsewhere, the methodology is very well described and straightforward and is appropriate for the questions being investigate.  The experimental data is suitable, accurately analyzed and the statistical analyses seem appropriate. 
  •  
  • The conclusions are exceptionally well stated, well thought-out and supported by their data. The paper is highly appropriate for the journal and the data are scientifically and technically sound, appear repeatable, and appropriately analyzed.

This paper will be of great interest to climate scientists, biocontrol scientists, ecologists and entomologists and especially those interested in effects of temperature stress on insects and parasitism.  The overall merit of the paper is significant. 

The paper is well written but there are a few grammatical corrections that should be made by the authors.  Please see the marginalia comments in the attached manuscript. 

Accept with minor corrections.

Please see attached comments in the manuscript.

Reviewer 2 Report

The present study deals with effect of short-term high temperature exposure of parasitoid wasp, A.asychis on survival and antioxidative defense system. Research seems reasonable in regard to importance of this parasitoid wasp in biological control of aphids in greenhouses. The manuscript is well-written and seems sound. It's clear that this study is part of great transcriptome study, but we don't have any details about that.

There are some minor mistakes which should be corrected:

Abstract:

The flow of the story in Abstract is not good. It has to make some reorganization, for example like that:

In this study, survival rate and anatioxidative enzymes pattern under short-term high temperature exposure of parasitoid wasp, A.asychis were examined. It is well known that high temperature, as typical negative environmental factor reduce the parasitism of this parasitoid.  Generally high temperature causes the rapid overproduction of reactive oxygen species (ROS) in organisms and antioxidative enzymes participate in the process of resisting environmental stress by eliminating excess ROS.  In present study, we identified two superoxide dismutase (SOD) ....

Introduction:

Line 49-50 – in sentence “information transmission substance” change with “parts of cell signaling”

Line 57 – term “superoxide silver ions” is not correct. Did you mean superoxide ion radical?

Line 58 – it’s not “singlet oxygen” product of this reaction than molecular oxygen

Line 60 – delete “and superoxidation” in sentence, because it’s incorrect term

Line 81- “parasitic wasps” change with “parasitoid waps”

Materials and Methods

Line 127-128 - mention that results are included in Supplement

Line 143 – “mechanical replicates” change with “technical replicates”

Line 149- “..and the method of use” fill with” sample preparation and the method of use”

Results

Line 161 – delete (P<0.001) because it’s not correct (in Figure 1 is P< 0.05) and it’s not necessary to be here this information

Line 172 – in Figure legend change “*”, there is” with “Asterisk (*) indicate”

Line 176 – “Kang et al., unpublished data” doesn’t mean anything because there is nothing in Reference list or some link to any data.  How to find this reference after publishing?

Line 185 – in sentence “…are distributed in organelles such as mitochondria and nucleus” to include that AasySOD1 is extracellular isoform

Figure 2 – purple background is not appropriate to red letters

Line 211-212 – “except for AasyPOD4, AasyPOD5 …” do you mean “AasyPOD5, AasyPOD6”

Line 222 – Figure 3 in Figure legends instead dot put comma and lowercase case for “And”

Line 237- Figure 4 in Figure legends same comment as above

Discussion

Line 263-264 – reference [41] in line 264 move before comma in line 263

Line 266 – in sentence “…and protein peroxidation” change to “and protein oxidation” because it’s more appropriate term

Line 274 – in sentence “It is the most important substance” change to “It is the most important enzyme”

In Paragraph 305-311 – Do you have some comment about  AasyGST4 that have specific pattern  for male and very high level of expression for female, also this is only nuclear isoform?

Line 328 – link www.mdpi.com/xxx/s1,  doesn't work

The present study deals with effect of short-term high temperature exposure of parasitoid wasp, A.asychis on survival and antioxidative defense system. Research seems reasonable in regard to importance of this parasitoid wasp in biological control of aphids in greenhouses. The manuscript is well-written and seems sound. It's clear that this study is part of great transcriptome study, but we don't have any details about that.

There are some minor mistakes which should be corrected:

Abstract:

The flow of the story in Abstract is not good. It has to make some reorganization, for example like that:

In this study, survival rate and anatioxidative enzymes pattern under short-term high temperature exposure of parasitoid wasp, A.asychis were examined. It is well known that high temperature, as typical negative environmental factor reduce the parasitism of this parasitoid.  Generally high temperature causes the rapid overproduction of reactive oxygen species (ROS) in organisms and antioxidative enzymes participate in the process of resisting environmental stress by eliminating excess ROS.  In present study, we identified two superoxide dismutase (SOD) ....

Introduction:

Line 49-50 – in sentence “information transmission substance” change with “parts of cell signaling”

Line 57 – term “superoxide silver ions” is not correct. Did you mean superoxide ion radical?

Line 58 – it’s not “singlet oxygen” product of this reaction than molecular oxygen

Line 60 – delete “and superoxidation” in sentence, because it’s incorrect term

Line 81- “parasitic wasps” change with “parasitoid waps”

Materials and Methods

Line 127-128 - mention that results are included in Supplement

Line 143 – “mechanical replicates” change with “technical replicates”

Line 149- “..and the method of use” fill with” sample preparation and the method of use”

Results

Line 161 – delete (P<0.001) because it’s not correct (in Figure 1 is P< 0.05) and it’s not necessary to be here this information

Line 172 – in Figure legend change “*”, there is” with “Asterisk (*) indicate”

Line 176 – “Kang et al., unpublished data” doesn’t mean anything because there is nothing in Reference list or some link to any data.  How to find this reference after publishing?

Line 185 – in sentence “…are distributed in organelles such as mitochondria and nucleus” to include that AasySOD1 is extracellular isoform

Figure 2 – purple background is not appropriate to red letters

Line 211-212 – “except for AasyPOD4, AasyPOD5 …” do you mean “AasyPOD5, AasyPOD6”

Line 222 – Figure 3 in Figure legends instead dot put comma and lowercase case for “And”

Line 237- Figure 4 in Figure legends same comment as above

Discussion

Line 263-264 – reference [41] in line 264 move before comma in line 263

Line 266 – in sentence “…and protein peroxidation” change to “and protein oxidation” because it’s more appropriate term

Line 274 – in sentence “It is the most important substance” change to “It is the most important enzyme”

In Paragraph 305-311 – Do you have some comment about  AasyGST4 that have specific pattern  for male and very high level of expression for female, also this is only nuclear isoform?

Line 328 – link www.mdpi.com/xxx/s1,  doesn't work

Author Response

Response to Reviewer 2 Comments

Thank you for taking time out of your busy schedule to review the manuscript. Now we have carefully corrected and replied the manuscript for this revision. The revision instructions are as follows:

Point 1:Abstract:

The flow of the story in Abstract is not good. It has to make some reorganization, for example like that:

In this study, survival rate and antioxidative enzymes pattern under short-term high temperature exposure of parasitoid wasp, A.asychis were examined. It is well known that high temperature, as typical negative environmental factor reduce the parasitism of this parasitoid.  Generally high temperature causes the rapid overproduction of reactive oxygen species (ROS) in organisms and antioxidative enzymes participate in the process of resisting environmental stress by eliminating excess ROS.  In present study, we identified two superoxide dismutase (SOD) ....

Response 1: Thanks for your valuable comment. Abstract, Line 29-35–The content of this section has been changed to “It is well known that high temperature, a typical negative environmental factor, reduces the parasitism of parasitoid. Generally, high temperature causes the rapid overproduction of reactive oxygen species (ROS) in organisms and antioxidative enzymes participate in the process of resisting environmental stress by eliminating excess ROS. In present study, we identified two superoxide dismutase (SOD), one catalase (CAT), six peroxidases (POD) and five glutathione-S-transferase (GST) genes, and survival rate and antioxidative enzymes pattern under short-term high temperature exposure of parasitoid wasp, A.asychis, were examined.”

Point 2: Introduction: Line 49-50 – in sentence “information transmission substance” change with “parts of cell signaling”

Response 2: Thanks for your valuable comment. Line 53-54–This error has been corrected according to you requested.

Point 3: Introduction: Line 57 – term “superoxide silver ions” is not correct. Did you mean superoxide ion radical?

Response 3: Thanks for your valuable comment. Line 61–The term “superoxide silver ions” should be modified to “superoxide anions”.

Point 4: Introduction: Line 58 – it’s not “singlet oxygen” product of this reaction than molecular oxygen

Response 4: Thanks for your valuable comment. Line 62–This error has been corrected according to you requested.

Point 5: Introduction: Line 60 – delete “and superoxidation” in sentence, because it’s incorrect term

Response 5: Thanks for your valuable comments. Line 64–“and superoxidation” has been removed at your request.

Point 6: Introduction: Line 81 – “parasitic wasps” change with “parasitoid waps”

Response 6: Thanks for your valuable comments. Line 85–"parasitic wasps" has been modified to "parasitoid waps" as you requested.

Point 7: Materials and Methods: Line 127-128 - mention that results are included in Supplement

Response 7: Sorry, I didn't understand your comment. Please give me more detailed information. If you are referring to the unmodified phylogenetic tree, we do not think that Figure 2 affects the presentation of the original data.

Point 8: Materials and Methods: Line 143 – “mechanical replicates” change with “technical replicates”

Response 8: Thanks for your valuable comments. Line 152–”mechanical replicates” has been modified to "technical replicates" as you requested.

Point 9: Line 149- “..and the method of use” fill with” sample preparation and the method of use”

Response 9: Thanks for your valuable comments. Line 160-161–”... and the method of use” has been replaced with "sample preparation and the method of use" as you requested.

Point 10: Results: Line 161–delete (P<0.001) because it’s not correct (in Figure 1 is P< 0.05) and it’s not necessary to be here this information

Response 10: Thanks for your valuable comments. Line 172–This error has been removed as you requested.

Point 11: Line 172 – in Figure legend change “*”, there is” with “Asterisk (*) indicate”

Response 11: Thanks for your valuable comments. Line 182–”*”, there is” has been replaced with "Asterisk (*) indicate" as you requested.

Point 12: Line 176 – “Kang et al., unpublished data” doesn’t mean anything because there is nothing in Reference list or some link to any data.  How to find this reference after publishing?

Response 12: Thanks for your valuable comments. Line 186–“Kang et al., unpublished data” has been removed. Since I do not have the authority to decide whether or not to publish this part of the data, I cannot explain further.

Point 13: Line 185 – in sentence “…are distributed in organelles such as mitochondria and nucleus” to include that AasySOD1 is extracellular isoform

Response 13: Thanks for your valuable comments. Line 195-196–We supplemented the subcellular location predictions for AasySOD1 and AasyPOD6.

Point 14: Figure 2 – purple background is not appropriate to red letters

Response 14: Thanks for your valuable comments. Figure 2–Line 211-213–The purple background in Figure 2 has been changed to light blue for better presentation of the information.

Point 15: Line 211-212 – “except for AasyPOD4AasyPOD5 …” do you mean “AasyPOD5, AasyPOD6

Response 15: Thanks for your valuable comment. Line 223–“except for AasyPOD4AasyPOD5 …” has been changed to “AasyPOD5AasyPOD6

Point 16: Line 222 – Figure 3 in Figure legends instead dot put comma and lowercase case for “And”

Response 16: Thanks for your valuable comment. Line 234–“And” have been replaced by comma.

Point 17: Line 237- Figure 4 in Figure legends same comment as above

Response 17: Thanks for your valuable comment. Line 249–“And” have been replaced by comma.

Discussion

Point 18: Line 263-264 – reference [41] in line 264 move before comma in line 263

Response 18: Thanks for your valuable comment. Line 275–reference [41] has been moved behind “... are larger than males”

Point 19: Line 266 – in sentence “…and protein peroxidation” change to “and protein oxidation” because it’s more appropriate term

Response 19: Thanks for your valuable comment. Line 279–”…and protein peroxidation” has been modified to "and protein oxidation" as you requested.

Point 20: Line 274 – in sentence “It is the most important substance” change to “It is the most important enzyme”

Response 20: Thanks for your valuable comment. Line 286– ”It is the most important substance” has been modified to "It is the most important enzyme" as you requested.

Point 21: In Paragraph 305-311 – Do you have some comment about  AasyGST4 that have specific pattern  for male and very high level of expression for female, also this is only nuclear isoform?

Response 21: Thanks for your valuable comment. Line 321-324–We've added the following comments on  AasyGST4.

The expression level of AasyGST4 in females increased significantly with treatment temperature, while it was stable in males. Due to the lack of genome, we are not sure about whether AasyGST4 is a nuclear isomer or not. We hypothesized that in males, AasyGST4 do not seem to be involved in the response to high temperature stress.

Point 22: Line 328 – link www.mdpi.com/xxx/s1,  doesn't work

Response 22: Thanks for your valuable comment. I'll contact the editor to fix this problem.

Thank you again for your suggestions. According to your suggestions, we have made corresponding changes and adjustments in the manuscript. The amendment to the question chapter can be found in the new version.

Reviewer 3 Report

In this manuscript authors have evaluated the expression levels of antioxidant genes and enzymatic activities in the parasitoid Aphelinus asychis at increasing temperatures. Overall, the manuscript is well written, materials and methods are complete and detailed, results are interesting. I have the following comments:

-L107-108: Is each replicate composed by 100 individuals (as written in L101)?

-L121: replace with MAFFT

-L132-133: at least, few relevant information on how insects were processed must be added, e.g., were insects full-body extracted? was liquid nitrogen added for grinding? Which DNAse procedure was used?

-L142: according to MIQE guidelines it should be RT-qPCR (as written in L208)

-L143: replace “mechanical” with “technical”

L152-153: It is not reported how raw data from RT-qPCR were processed before the analysis. The two commonest ways are delta-delta-Ct (Livak and Schmittgen, 2001, doi: 10.1006/meth.2001.1262) and relative quantification (Pfaffl, 2001, doi: 10.1093/nar/29.9.e45), and subsequent modifications to include, e.g., inter-run calibrations. Additionally, it is not clear whether the reference gene was evaluated for every RT-qPCR run or only once. If only once, it is crucial that authors specify that RT-qPCR were conducted using a sample maximization procedure rather than a gene maximization procedure, or maybe a mix of the two but inter-run calibrators should had been used. There are several examples of papers that adopted such approach, this  is clearly discussed in Hellemans J, Vandesompele J (2011) Quantitative PCR data analysis–unlocking the secret to successful results. In: Kennedy S, Oswald N (eds) PCR troubleshooting and optimization. The essential guide. Caister Academic Press, Norfolk, UK, pp 1–13

L170: in figure caption it would be useful to better clarify that UPPERCASE letters are for females and lowecase letters are for males – as written it seems the opposite

L183: please, replace “Major” with “The majority” (if I am correctly interpreting the text)

L203-204: please specify in caption what is the meaning of Sigma, Theta, Delta

Author Response

Response to Reviewer 3 Comments

Thank you for taking time out of your busy schedule to review the manuscript. Now we have carefully corrected and replied the manuscript for this revision. The revision instructions are as follows:

Point 1: -L107-108: Is each replicate composed by 100 individuals (as written in L101)?

Response 1: Thanks for your valuable comment. Each replicate contained 100 individuals, but only those individuals that survived treatment were used for extraction, so each replicate used a different number of individuals for RNA extraction. 100 individuals per replicate ensured that the number of individuals surviving the highest temperature (41°C) treatment also extracted a sufficient amount of RNA.

Point 2: -L132-133: at least, few relevant information on how insects were processed must be added, e.g., were insects full-body extracted? was liquid nitrogen added for grinding? Which DNAse procedure was used?

Response 2: Thanks for your valuable comment. Line 127– “MAFF” has been replaced with “MAFFT”.

Point 3: -L132-133: at least, few relevant information on how insects were processed must be added, e.g., were insects full-body extracted? was liquid nitrogen added for grinding? Which DNAse procedure was used?

Response 3: Thanks for your valuable comment. Line 111–Whole individuals were used for RNA extraction. Line 111–The grinding process of the samples was described. Line 148-151–The system and cycling conditions for RT-qRCR are described.

Point 4: -L142: according to MIQE guidelines it should be RT-qPCR (as written in L208)

Response 4: Thanks for your valuable comment. Line 151–”qRT-PCR” has been corrected to “RT-qPCR”.

Point 5: -L143: replace “mechanical” with “technical”

Response 5: Thanks for your valuable comment. Line 152–”mechanical” has been corrected to “technical”.

Point 6: L152-153: It is not reported how raw data from RT-qPCR were processed before the analysis. The two commonest ways are delta-delta-Ct (Livak and Schmittgen, 2001, doi: 10.1006/meth.2001.1262) and relative quantification (Pfaffl, 2001, doi: 10.1093/nar/29.9.e45), and subsequent modifications to include, e.g., inter-run calibrations. Additionally, it is not clear whether the reference gene was evaluated for every RT-qPCR run or only once. If only once, it is crucial that authors specify that RT-qPCR were conducted using a sample maximization procedure rather than a gene maximization procedure, or maybe a mix of the two but inter-run calibrators should had been used. There are several examples of papers that adopted such approach, this  is clearly discussed in Hellemans J, Vandesompele J (2011) Quantitative PCR data analysis–unlocking the secret to successful results. In: Kennedy S, Oswald N (eds) PCR troubleshooting and optimization. The essential guide. Caister Academic Press, Norfolk, UK, pp 1–13

Response 6: Thanks for your valuable comment. Line 152–The processing method of RT-qPCR raw data is described. Line 156–The reference gene was evaluated in eacg RT-qPCR, and we have added a statement to it.

Point 7: L170: in figure caption it would be useful to better clarify that UPPERCASE letters are for females and lowercase letters are for males – as written it seems the opposite

Response 7: Thanks for your valuable comment. Line 181, Line 232-234, Line 247-249–These errors have been corrected.

Point 8: L183: please, replace “Major” with “The majority” (if I am correctly interpreting the text)

Response 8: Thanks for your valuable comment. Line 194–“Major” has been corrected to “The majority”

Point 9: L203-204: please specify in caption what is the meaning of Sigma, Theta, Delta

Response 9: Thanks for your valuable comment. Line 206-207, Line 210-213–Sigma, Theta, Delta represent the different classes. In order not to cause misunderstanding, we have changed Sigma, Theta, Delta to Class I, Class II, Class III in the manuscript and pictures.

Thank you again for your suggestions. According to your suggestions, we have made corresponding changes and adjustments in the manuscript. The amendment to the question chapter can be found in the new version.

Round 2

Reviewer 3 Report

The authors have addressed all relevant comments. In my opinion the manuscript can be published.